# Response of Benthic Diatom Assemblages to Contamination by Metals in a Marine Environment

**Yuriko Jocselin Martínez** [1,*]**, David Alfaro Siqueiros-Beltrones** [1,*] **and Ana Judith Marmolejo-Rodríguez** [2,*]

1 Departamento de Plancton y Ecología Marina, Centro Interdisciplinario de Ciencias Marinas—Instituto Politécnico Nacional. Av. IPN s/n, Col. Playa Palo de Santa Rita, La Paz CP 23096, Mexico
2 Departamento de Oceanología, Centro Interdisciplinario de Ciencias Marinas—Instituto Politécnico Nacional, Av. IPN s/n, Col. Playa Palo de Santa Rita, La Paz CP 23096, Mexico
\* Correspondence: okiruy20g@gmail.com (Y.J.M.); dsiquei@gmail.com (D.A.S.-B.); amarmole@ipn.mx (A.J.M.-R.)

**Abstract:** Studies on marine benthic diatoms in environments contaminated by metals are scarce. The typical structure of benthic diatom assemblages (species richness, diversity, dominance, dominant taxa) from undisturbed environments may be used as reference for contrasting with contaminated environments in order to observe how said assemblages respond to such disturbance. Thus, the Ho that the structure of benthic diatom associations and morphology of their frustules under contamination by metals would be normal, as in unpolluted environments was tested. To do this, concentrations of 24 metals were surveyed in a coastal zone impacted by mining residues, and the structure of benthic local diatom assemblages was described. Metal concentrations measurements for 15 metals surpassed the normal values of the upper earth cortex, seven were under the low range effect, and three (Cd, Cu, Zn) surpassed the medium range effect values. At a control site no element concentration was above the reference values for low range effect (LRE) or medium range effect (MRE) standards. There, diatom species richness (S) was high, particularly on seaweeds; where, 397 diatom taxa were recorded. In contrast, at the contaminated area 217 diatom taxa were recorded, but diversity (H') ranged from 2.4 to 4.3. Relative high frequencies of deformed diatom valves mainly of *Achnanthes* spp. were recorded in contaminated sediments. In general, diatom taxocenoses presented a typical structure for non-contaminated environments. However, scarceness of specimens, lower S, and frequency of deformed valves suggest responses to metal contamination. For marine environments, the latter values corresponding to *A. longipes* may be considered a reliable reference to the response of benthic diatoms to metal contamination.

**Keywords:** marine; diatoms; metals; pollution; teratology

## 1. Introduction

Marine environments are subject to different types of pollution due to anthropic activities, either due to input of excess organic matter or metals, that affect the habitat and biota. This has led to the search for taxa that may serve as bio-indicators of the various types of pollution. In any case, those taxa having benthic habits, short reproductive cycles, and that are sensible or tolerant to contaminants are preferred. In this way, indicator species of polychaetes have been identified, e.g., *Capitella capitata* for organic matter pollution, and *Nereis diversicolor* for high concentration of metals [1,2]. In the case of microalgae, the observed responses of diatoms suggest that certain taxa have resistance or tolerance to contamination by metals [3,4], such as *Achnanthes minutissima* and *Navicula seminulum* (*N. atomus*) that in continental habitats can thrive in the presence of various metals. Specifically, *A. minutissima* and *Brachysira vitrea* have been recorded as dominant species under the influence of Zn, Cd and Fe, which reflects their opportunistic strategies and tolerance to Zn and Cd [5], but deformed valves with distorted axial areas of *Fragilaria* cf. *tenera*, *Fragilaria capucina* var. *rumpens* and *Eunotia* sp. are also frequent in continental environments [6].

According to this teratological approach, in marine habitats contaminated with metals Dickman [7] recorded deformed frustules of *Fragilaria capucina, Achnanthes hauckiana, Diatoma vulgare, Navicula rhynchocephala*, although these taxa are typically regarded as freshwater forms. Moreover, deformed frustules of benthic marine diatoms are not rare albeit they occur in low frequencies. In a study comprising three distinct environments contaminated with metals, freshwater, brackish, and marine, less than 0.5% of deformed valves were recorded in marine diatom assemblages [8].

Besides the above, physiological responses of benthic diatoms to contamination by metal are also an issue. In vitro studies using marine diatoms show their response to mean lethal concentrations (EC50). For example, the growth of *N closterium* at Zn, Cd, Cu, Pb concentrations of 0.065 mg $L^{-1}$, 79 mg $kg^{-1}$, 26 mg $kg^{-1}$, 29 mg $kg^{-1}$ is inhibited [9,10]. This effect has been observed also in *Entomoneis* cf. *punctulata, Nitzschia* cf. *paleacea, Amphora coffeiformis, Odontella mobiliensis, Amphora hyalina, Thalassiosira nordenshkioelditii, Thalassiosira pseudonana*, and *Thalassiosira weissflogii*, with intracellular Cd accumulation [11–17].

Under a different perspective, the typical structure of benthic diatom assemblages (species richness, diversity, dominance, dominant taxa) may be used as reference for contrasting with contaminated environments. Several studies on benthic marine diatoms in non-polluted environments show that the species richness in a given substratum varies between 20 and 50 taxa per sample, depending on the type of substrate, harshness of the environment, and degree of habitat disturbance [3,18]. While species richness of benthic diatoms in (undisturbed) highly productive environments commonly range between 50 and 100 taxa per sample, reaching over 300 total taxa [19]. In contrast, under harsh or extreme conditions, e.g., hypersaline environments, species richness of benthic diatom may be very low in a sample, or overall, e.g., 6 to <20 taxa, respectively [3,4]. Notwithstanding, in general, benthic diatom assemblages exhibit a similar structure, i.e., few abundant and common taxa, and many rare and uncommon taxa [3] which has to be considered when measuring species diversity in an ecological perspective, including forcing by contamination.

Regarding species diversity, the interpretation of computed values under the information theory concept (H′) in benthic diatom assemblages has allowed to stablish empirically a typical variation interval of diversity values (H′ = 2.6–3.89) that reflects either unfavorable or favorable conditions on the basis of certainty that these represent real and thus stable conditions in which benthic diatom taxocenoses grow [4]. In this way, said interval can be used as reference to infer likely ecological impact due to changes in the habitat, whether by natural or anthropic activities in the various marine environments. Studies on the response of benthic diatom assemblages to contamination by metals in marine environments are scarce, and although these correspond with low values of diversity in the area [7,8,20–25], a reliable model recording the response of benthic diatoms to metal pollution is still lacking.

A recent exploratory study in the mining town Santa Rosalia, located on the western coast of the Gulf of California, Mexico by Martínez et al. (unpublished) represents the first ex professo investigation on this subject. This is an area contaminated by mining residues that, although having human settlements and where various contamination related studies have been done [25–29], no study had hitherto considered the potential response of benthic diatoms to the elevated concentrations of certain metals. Benthic diatom assemblages inspected by Martínez et al. (unpublished) showed an apparent high frequency of deformed frustules, suggesting that an impact by potentially toxic elements (PTE) derived from mining activities exists but, also, an ability of the diatoms to tolerate elevated concentrations of PTE such as metals, inasmuch most of the estimated values of the diatom assemblage parameters were within typical intervals. This, in spite of the medium range effect concentrations (MRE) of PTE measured in the study area, which implied that 50% of the biota present should be affected. All this suggests indeed that certain diatom taxa have an ability to resist or tolerate contamination by metals, inasmuch they persist in said polluted sites as suggested earlier by Siqueiros-Beltrones [3,4].

On the basis of the above and mainly on the aforementioned exploratory study, we set out to test the null hypothesis that the structure of the benthic diatom assemblages

and the frequency of occurrence of deformed valves would not differ from that of marine environments not contaminated by metals. Thus, the aim of this study was to determine how the association of the benthic marine diatoms living in the beach of Santa Rosalia are structured, and frequency of deformed valves, under contamination by metals derived from mining activities, and to contrast it with an unpolluted control site.

## 2. Study Area

Santa Rosalia (SR) mining town is located (27°20′ N, 112°16′ O) in the northern part of Baja California Sur (BCS), Mexico facing the Gulf of California [30]. Annual pp is quite low (<100 mm) in the area and mainly during short periods associated to tropical storms [31]. It is influenced by Gulf of California currents (GCC) which are linked to the Mexican Coastal Current intensification. These interact in a way that the lateral sheer between them generates eddies in the gulf entrance. The GCC flows S-N with alternating fluxes by entering along the Eastern coast, and exits (N-S) along the peninsula coast [32,33]. This was taken into consideration for choosing the control site (CS) in order to avoid contamination by metals from the SR mining activities. Our CS was thus located 8 km northward of SR in the locality known as Santa Maria (Figure 1).

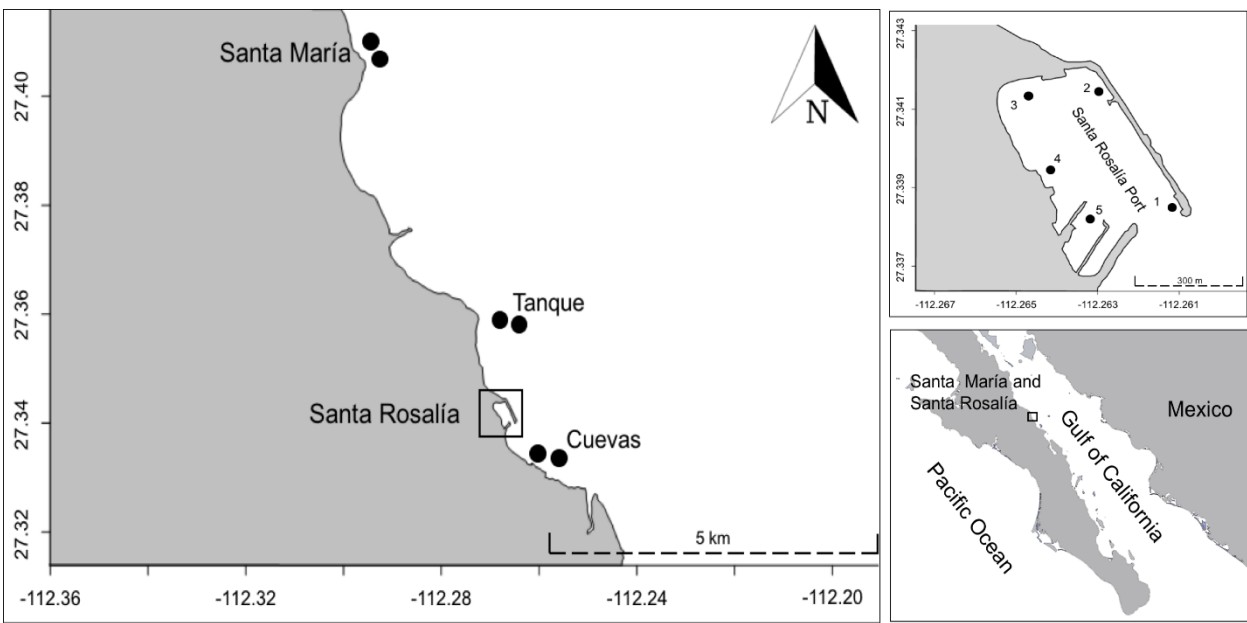

**Figure 1.** Location of study area (Santa Rosalia) and control site (Santa María).

A biogeochemical study off the coast of Santa Rosalia by Shumillin et al. (2012) detected that Cd concentration levels in sediments were slightly enriched, albeit under the low range effect (LRE) standard, or that it affects 10% of the biota. Whereas Pb, Cu (3390 ± 804 mg/kg) and Zn (1916 ± 749 mg/kg) concentrations in most of the sediment samples were higher than the medium range effect (MRE), which implies that 50% of the biota may be affected, thus indicating an extensive anthropogenic impact by this metal due to mining activities in beach sediments and indicating a high toxicological hazard for the resident marine biota of the area. We thus assumed that whatever extreme-like responses diatoms may exhibit would most likely be attributed to the high metal concentrations in the area.

## 3. Materials and Methods

Sampling design relied mainly on observations in the exploratory study [34], inasmuch no specific background on the topic was available for the study area. For the purposes of this study Santa Rosalia is considered to comprise two zones: Port with its respective dock,

and the coast where the northern part is known as Punta Tanques and the southern part is Cuevas (Figure 1).

In the Port zone two rocks (2–3 kg) and eight sediment samples were collected during May 2015 and January 2016. Moreover, sampling was done in two stations at Tanques and Cuevas during March 2016 (Table 1). Sampled substrates included macroalgae thallii and sediments at a distance of 20 m from the coast and a depth of 8–10 m. A single thallus of each collected macroalgal taxon (genus) was deposited in plastic (Ziploc) bags. Sediments were collected using a previously washed Petri dish and used for both diatom and geochemical analysis. Moreover, sediments and macroalgae thalli were collected in two points at the control site. The control site (CS) was located 8 km northward of SR in the locality known as Santa Maria in order to avoid any metal pollution from the SR mining activities. There, pristine conditions ensure that the benthic diatom assemblages used as reference would warrant high species richness and a species composition related to previous surveys in the gulf region [3,35] and those underway at the time [34,36].

**Table 1.** Sampling stations and sampled substrata in the study area and control site.

| | Study Area (Santa Rosalia) | | | Control Site (Santa Maria) | |
|---|---|---|---|---|---|
| **Station** | **Port** | **Costa Tanques** | **Costa Cuevas** | **1** | **2** |
| Sampling points | 5 | 2 | 2 | | |
| Substrate marine | Sediment Rock | Sediment Algae: *Sargassum* sp. *Dyctiota* sp. *Jania* sp. *Padina* sp. | Sediment Algae: *Sargassum* sp. *Dyctiota* sp. Alga 1. | Sediment Algae: *Codium* sp. *Sargassum* sp. Alga 1 | Sediment Algae: *Codium* sp. *Gracilaria* sp. *Sargassum* sp. |

### 3.1. Diatom Mounts

Sediment samples were obtained or separated in two subsamples: (a) for diatom analyses, and (b) trace elements. To separate diatoms, sediment was treated with ultrasound for detaching diatoms from the grains, and the decanted liquid was separated into a test tube and let to settle for 24 h. Epiphytic diatoms were scrapped off from the macroalgae thallii while rinsing with purified water, and the scrapped off material was poured into a test tube. In the case of epilithic forms, in each sampled rock an area of approximately 100 cm$^2$ was brushed using a toot-brush while rinsing with purified water, and the brushed-off material was poured into a test tube. All diatoms samples were placed in 60 mL test tubes and treated following Siqueiros-Beltrones [3], for oxidizing all organic matter inside and outside the diatom frustules in a mixture of sample, commercial alcohol, and nitric acid at a 1:1:5 ratio, rinsing repeatedly afterwards with purified water until reaching a pH > 6. Cleaned diatoms were mounted by duplicate on permanent slides in Pleurax (RI = 1.7).

Diatom taxa were identified under an Olympus compound microscope with planachromatic lens, phase contrast, and a photographic ocular lens at 1000 ×. Identification followed: Peragallo and Peragallo [37], Schmidt et al. [38], Hustedt [39], Round et al. [40], Witkowski et al. [41], Moreno et al. [35], Siqueiros-Beltrones [3,42], Hernández-Almeida and Siqueiros-Beltrones [43,44], López-Fuerte et al. [45], Siqueiros-Beltrones et al. [46].

According to Siqueiros Beltrones [3], the relative abundances (reference N = 500) of the identified species were estimated per sample and used to compute the association parameters with the following indices: species diversity (Shannon's H′ and Simpson's 1-*L*), equitability (Pielou's J′), dominance (Simpson's *L*). The diversity values from the derived matrix were compared for significance differences among the Puerto, and Costa (Tanques and Cuevas, SR), and CS stations, using a Kruskal–Wallis (one way) test for the Ho = no significant differences would be detected. The Bray–Curtis index was used as a measure of

difference between assemblages or samples inasmuch it relies on both presence/absence of taxa and their relative abundances. Clusters were derived from the Bray–Curtis similarity matrix of values using program Primer 6.

*3.2. Measurements of Metal Concentration*

Sediment samples were placed in labeled plastic containers and dried in a wooden oven at <50 °C for a week. Afterwards, the samples were pulverized using an agate mortar and placed in plastic vials. These were processed as follows: 0.25 g of the pulverized material were digested with four acids: HF, $HClO_4$, $HNO_3$, and HCl. Metal concentrations were then measured with an inductively coupled mass spectrometer (ICP-MS), using marine sediment certified reference standards PACS-2 and MESS-3 for technique validation. The measured elements include the following: Al (%), Fe (%), Mg (%), Hg ($\mu g\ kg^{-1}$), and mg $kg^{-1}$ (Ag, As, Ba, Bi, Cd, Co, Cr, Cu, In, Mn, Mo, Ni, Pb, Sb, Se, Sn, Sr, U, V, Zn). Validation of methods for the sediments are presented in Table 2.

**Table 2.** Reference standards PACS-2, MESS-3 (sediment), for the analyzed elements with the certified value, determined value, and percentage of recovery.

| Metal | Reference Standard MESS-3 | | | Reference Standard PACS-2 | | |
|---|---|---|---|---|---|---|
| | Certified Value | Determined Value | % Rec. | Certified Value | Determined Value | % Rec. |
| Al (%) | 8.59 | 7.48 | 87.08 | 6.62 | 5.96 | 90.03 |
| Ag | 0.18 | 0.27 | 150 | 1.22 | 1.24 | 101.64 |
| As | 21.2 | 21.9 | 103.3 | 26.2 | 26.5 | 101.15 |
| Ba | - | 1060 | - | - | 73 | - |
| Bi | - | 0.38 | - | - | 0.38 | - |
| Cd | 0.24 | 0.2 | 83.33 | 2.11 | 2.5 | 118.48 |
| Co | 14.4 | 12.7 | 88.19 | 11.5 | 10.8 | 93.91 |
| Cr | 105 | 84.9 | 80.86 | 90.7 | 66.7 | 73.54 |
| Cu | 33.9 | 35.4 | 104.42 | 310 | 328 | 105.81 |
| Fe (%) | 4.34 | 4.11 | 94.70 | 4.09 | 4 | 97.80 |
| Hg | 0.091 | 0.12 | 131 | 3.4 | 2.55 | 75 |
| In | - | 0.1 | - | - | 0.3 | - |
| Mg (%) | 1.6 | 1.52 | 95.00 | 1.47 | 1.27 | 86.39 |
| Mn | 324 | 329 | 101.54 | 440 | 427 | 97.05 |
| Mo | 2.78 | 2.76 | 99.28 | 5.43 | 5.9 | 108.66 |
| Ni | 46.9 | 38.2 | 81.45 | 39.5 | 23 | 58.23 |
| Pb | 21.1 | 22 | 104.27 | 183 | 157 | 85.79 |
| Sb | 1.02 | 0.4 | 39.22 | 11.3 | 6 | 53.1 |
| Se | 0.72 | 0.3 | 41.67 | 0.92 | 0.7 | 76.09 |
| Sn | 2.5 | 2 | 80 | 19.8 | 21 | 106.06 |
| Sr | 129 | 133 | 103.1 | 276 | 271 | 98.19 |
| U | 4 | 4 | 100 | 3 | 2.6 | 86.67 |
| V | 243 | 211 | 86.83 | 133 | 119 | 89.47 |
| Zn | 159 | 151 | 94.97 | 364 | 400 | 109.89 |

Units for elements are mg $kg^{-1}$, except Hg which is in $\mu g\ kg^{-1}$.

To calculate the normalized enrichment factor (NEF) the concentration values of the elements were normalized with Al using the formula: $FEN_M = (M_{SAMPLE}/Al_{SAMPLE})/(M_{CTS}/Al_{CTS})$; where M is a metal and CTS corresponds to the values of the upper continental crust [47]. The estimated metal concentrations were compared with the values of toxicity proposed by Long et al. [48], which define the LRE and MRE. The FEN values allowed estimate sediment quality or degree of contamination according to a scale for mining stations: 1–3 = low contamination; 3–10 = moderate; 10–25 = severe; 25–50 = very severe; and >50 = extremely severe contamination [49].

## 4. Results

### 4.1. Metal Concentrations

Average values of metal concentrations from the sampling stations in SR and CS were acquired (Table 3). These were contrasted with values for the upper continental crust (UCC) mean values [47], the reference values for a polluted environment, and the LRE and MRE values reported by Long et al. [48]. Out of the 18 metals surveyed, 15 had values higher than the UCC values. The values for six of these metals surpass the LRE, and three are higher than the MRE in SR, i.e., Zn (4755, 740.5 mg kg$^{-1}$), Cd (42.31 mg kg$^{-1}$) and Cu (3193 mg kg$^{-1}$) surpassing the established limits (Cd = 9.6, Zn = 410, Cu = 270 mg kg$^{-1}$), and implying that 50% of the biota may be affected by their presence at these concentration levels. In particular, in the SR port, Ni reached a concentration of 401 mg kg$^{-1}$, which is over the MRE (51.6 mg kg$^{-1}$) limit, while all the other sampling stations had values below 0.5 mg kg$^{-1}$. Moreover, Zn reached concentrations above 700 mg kg$^{-1}$, and up to 4030 mg kg$^{-1}$, way above the MRE (410 mg kg$^{-1}$) limit. Other outstanding concentrations were those of Cu, with values between 1223 y 7980 mg kg$^{-1}$, vs. the MRE limit of 270 mg kg$^{-1}$. These elements did not show elevated concentration at the CS, although they surpassed the UCC values: Zn (119.9 mg kg$^{-1}$) vs. UCC = 52 mg kg$^{-1}$, Cu (31.9 mg kg$^{-1}$) vs. UCC = 14 mg kg$^{-1}$, and Sr (935.6 mg kg$^{-1}$) vs. UCC = 316 mg kg$^{-1}$, although no LRE values are available. As expected, no element in the CS was above the reference values for LRE or MRE standards.

**Table 3.** Metal concentrations in sediments for the sampling stations in Santa Rosalia-Baja California Sur, control site, and contamination references. P = port, C = Cuevas, T = tanques, CS = control site, UCC = Upper earth cortex value, LRE = Low range effect, MRE = Medium range effect (mg kg$^{-1}$).

|        | P1      | P2      | P3      | P4    | P5    | C      | T     | SC     | UCC   | LRE  | MRE  |
|--------|---------|---------|---------|-------|-------|--------|-------|--------|-------|------|------|
| Ag     | 0.5     | 0.9     | 0.6     | 0.5   | 0.4   | 0.53   | 0.5   | 0.12   | 0.1   | 1    | 3.7  |
| Al (%) | 3.3     | 5.4     | 6.6     | 7.4   | 7.9   | 6.73   | 8.55  | 5.8    | 7.74  | -    | -    |
| Ba     | 2290    | 660     | 4725    | 1915  | 1168  | >5000  | 1520  | 493.3  | 668   | -    | -    |
| Cd     | 1.8     | 3.2     | 2.2     | 1.8   | 1.7   | 2.2    | 243   | 0.3    | 0.1   | 1.2  | 9.6  |
| Co     | 301     | >500    | >500    | 231   | 103   | >500   | 109   | 8.96   | 11.6  | -    | -    |
| Cr     | 45      | 59.6    | 98.4    | 65    | 52.4  | 56.9   | 62    | 39.2   | 35    | 81   | 370  |
| Cu     | 3130    | 7980    | 4250    | 2220  | 1223  | 4350   | 1560  | 31.96  | 14.3  | 34   | 270  |
| Fe (%) | 4.6     | 6.3     | 9.2     | 5.7   | 4.6   | 7.425  | 5.11  | 2.9    | 3.08  | -    | -    |
| Hg     | 60      | 60      | 50      | 105   | 70    | 25     | 40    | 48.33  | 56    | 150  | 710  |
| Li     | 58      | 87.8    | 140.5   | 44.9  | 24.2  | 162.5  | 19.65 | 13.26  | 22    | -    | -    |
| Mn     | >10,000 | >10,000 | >10,000 | 7720  | 5245  | >10,000| 5465  | 744    | 527   | -    | -    |
| Mo     | 12.1    | 21.5    | 47.2    | 7.8   | 8.9   | 18.5   | 0.305 | 1.15   | 1.4   | -    | -    |
| Ni     | 401     | <0.5    | <0.5    | <0.5  | <0.5  | 16.1   | 52.9  | 0.5    | 18.6  | 20.9 | 51.6 |
| Pb     | 135     | 383     | 234     | 219   | 111   | 276.5  | 48.9  | 11.16  | 17    | 46.7 | 218  |
| Sn     | 4       | 15      | 6       | 5     | 8     | 3      | 2     | 1      | 2.5   | -    | -    |
| Sr     | 1490    | 1820    | 2555    | 1259  | 984   | 7550   | 458.4 | 935.6  | 316   | -    | -    |
| U      | 27.3    | 56.7    | 63.6    | 16.8  | 8     | 110.5  | 2.95  | 2      | 2     | -    | -    |
| V      | 107     | 162     | 194     | 106.5 | 94.5  | 177.5  | 79.5  | 66     | 53    | -    | -    |
| Zn     | 1990    | 3320    | 4030    | 1430  | 703   | 4755   | 740.5 | 119.9  | 52    | 150  | 410  |

### 4.2. Normalized Enrichment Factor (NEF)

In relation with the above observations, the NEF values were computed for metal concentrations in sediment samples of SR. For Cd, Mn, Co, Zi, Mo, Pb, U, Bi, and In, these were above the moderate level, corresponding with a NEF = 10, indicating a contamination level. The sequence of enriched elements was: Cu > Zn > In > Cd > Co > Mn > Bi > U > Pb > Mo (Figure 2). Points 4 and 5 of SR port measured concentrations surpass the MRE for Zn = 1430, 703, ERM = 410, Cu = 2220, 1223, ERM = 270, Pb = 219, 111, ERL = 46.7, ERM = 218, where a higher frequency of deformed diatom valves was recorded as well as a lower value of species diversity (H′ = <1.9).

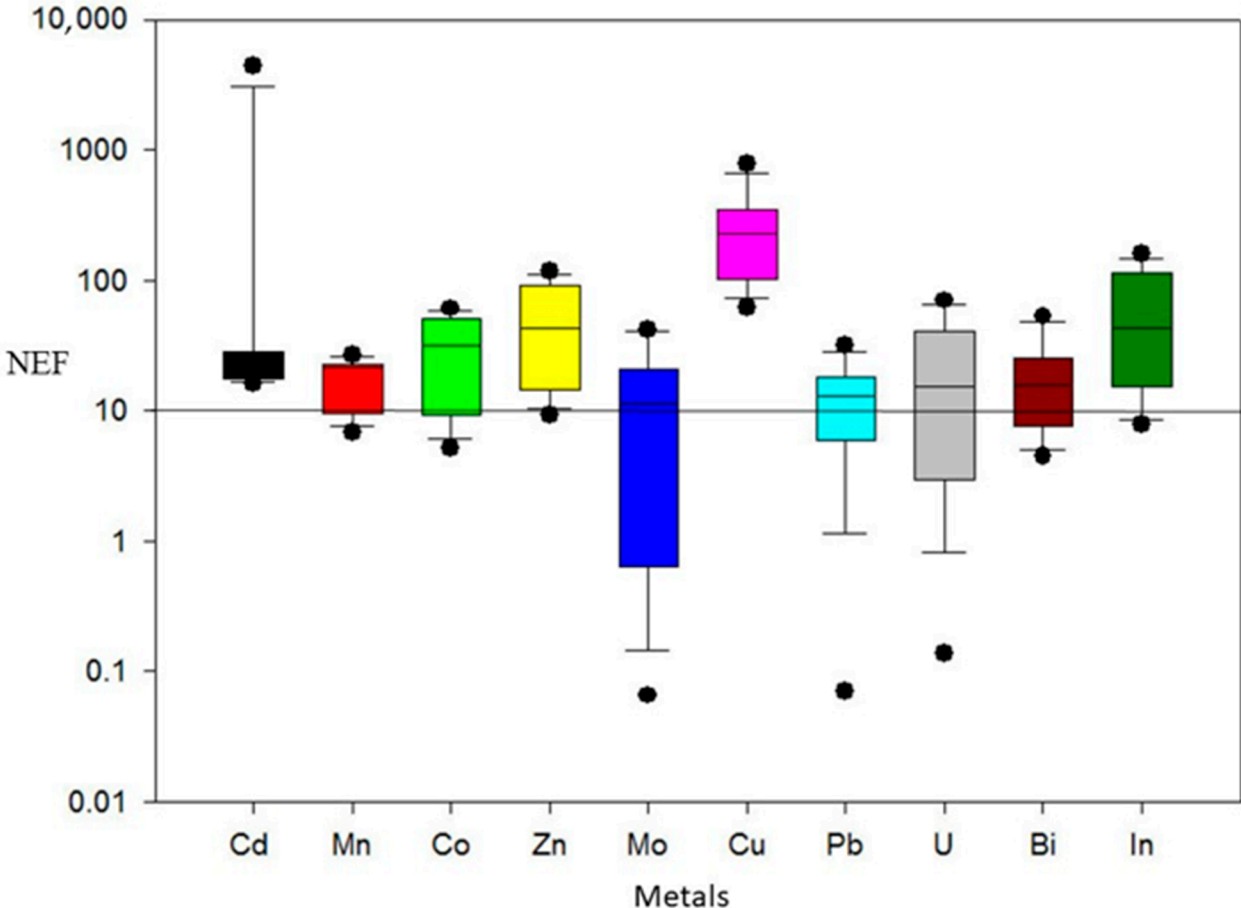

**Figure 2.** Selected metals showing enriched concentrations in sampling stations of Santa Rosalia, BCS. Cd SD = ±289.5, Mn SD = ±4.8, Co SD = ±22, Zn SD = ±13.6, Mo SD = ±13.1, Cu SD = ±71.4, Pb SD = ±10.7, U SD = ±17.6, Bi SD = ±26.9, In SD = ± 24.1.

### 4.3. Diatom Assemblages

Benthic diatom floristics of all sampled stations and substrata in SR comprised 215 taxa (species, varieties, forms) within 63 genera vs. the 397 taxa recorded at the CS. The best represented genera in SR were *Amphora* with 28 taxa, *Nitzschia* (25), *Achnanthes* (11), *Cocconeis* (19), *Navicula* (15) and *Diploneis* (14).

In the numerical analysis 9472 (N) diatom valves were counted, yielding abundances of 1439 and 1192 valves for *Cocconeis scutellum* and *Gomphoseptatum aestuarii*, respectively, being thus the two dominant species in the SR assemblage, and accounting for almost 30% of the overall diatom abundance (N). Thereafter, the most common taxa were *Amphora ocellata, Catenula adherens, Navicula diversistriata, N. subinflatoides* and *Staurophora salina*, while in the CS, *Licmophora flabellata* was the most numerous taxon present (Table 4). Among the 21 surveyed substrates in SR, *Navicula diversistriata* (12 substrates), *Cocconeis scutellum* (11) *Caloneis liber* (10) and *Diploneis litoralis* (10) showed the highest frequencies. Similarity analysis (Figure 3) shows that each station had a distinct species composition in terms of the common taxa. *L. flabellata* was not found in the SR sampling stations, while *G. aestuarii* did not occur in the CS or in the port of SR. In general, the port and coast of SR exhibited distinct diatom assemblages.

**Table 4.** Abundances of benthic diatoms (N = 9472) in all substrates at control site (CS), Coast (Tanques-Cuevas), and Port, SR.

| Species | CS | COAST | PORT |
|---|---|---|---|
| *Amphora ocellata* | 0 | 8 | 333 |
| *Achnanthes javanica* | 0 | 0 | 197 |
| *Catenula adherens* | 0 | 478 | 0 |
| *Cocconeis scutellum* | 256 | 1439 | 0 |
| *Gomphoseptatum aestuarii* | 0 | 1192 | 0 |
| *Licmophora flabellata* | 1431 | 0 | 0 |
| *Navicula diversistriata* | 70 | 503 | 6 |
| *Navicula subinflatoides* | 0 | 0 | 320 |
| *Staurophora salina* | 0 | 0 | 483 |
| *Psammodyction constrictum* | 0 | 22 | 270 |
| Total | 4026 | 5289 | 4183 |

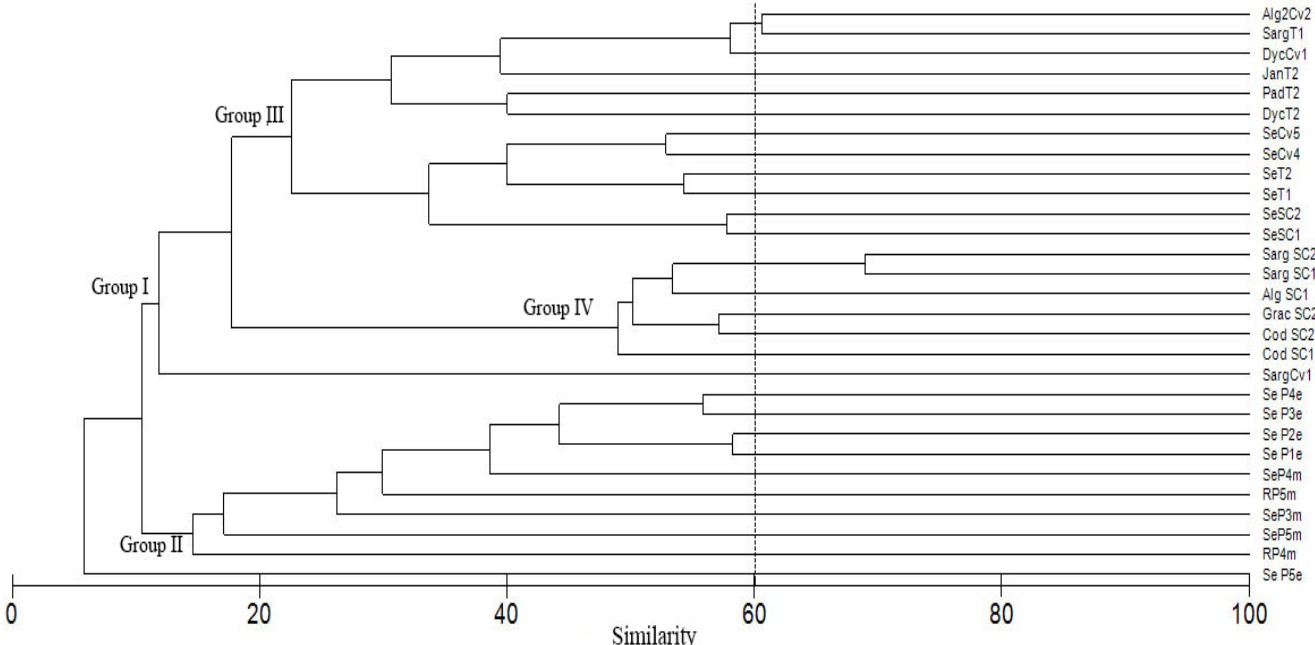

**Figure 3.** Bray Curtis similarity base done presence/absence data of the benthic diatom assemblages from Santa Rosalia and the control site. P = Puerto, T = Tanques, Cv = Cuevas, SC = control site, R = roca, Se = sediment, Jan = *Jania*, Sarg = *Sargassum*, Dyc = *Dyctiota*, Pad = *Padina*, Grac = *Gracilaria*, Cod = *Codium*.

### 4.4. Diatom Species Diversity

In the sediments of SR Port, diatom species richness (S) ranged from 6 to 47 taxa, with higher S at points 2 (41 taxa), 4 (45), 5 (47), while point 1 had only 6 taxa in January 2016; in May, 2015 sampling point 5 had 13 taxa, albeit N = 500, but in contrast with sampling point 3 during May, 2015 where only 169 valves were observed. In this station the lowest species diversity occurred at sampling point 5 during both months (H′ = 1.1 and 1.9); in other points values ranged from 2.4 to 4.3, with corresponding low dominance values (λ = 0.08–0.32). Exceptionally, on the rock collected at sampling point 4, only 14 valves from 8 taxa were observed, but with correspondent high diversity (H′ = 2.6) and dominance (λ = 0.13) values (Table 5).

**Table 5.** Index values describing the structure of benthic diatoms at sampling stations in Santa Rosalia and Control Site. Species richness (S), sample size (N), Pielou′s equitability (J′), Shannon′s diversity (H′), Simpson′s dominance (λ), Simpson′s diversity (1-λ). R = rock; Se = sediments, P = Port, T = Tanques, C = Cuevas, CS = control site, # = sampling point.

| Station | Substrate | S | N | J′ | H′ | λ | 1-λ |
|---|---|---|---|---|---|---|---|
| Port May 2015 | Rock P4 | 8 | 14 | 0.89 | 2.6 | 0.13 | 0.86 |
| | Rock P5 | 25 | 500 | 0.51 | 2.4 | 0.26 | 0.73 |
| | Se P3 | 30 | 169 | 0.75 | 3.6 | 0.14 | 0.85 |
| | Se P4 | 34 | 500 | 0.80 | 4.1 | 0.08 | 0.91 |
| | Se P5 | 13 | 500 | 0.53 | 1.9 | 0.32 | 0.67 |
| Port January 2016 | Se P1 | 47 | 500 | 0.77 | 4.3 | 0.07 | 0.92 |
| | Se P2 | 32 | 500 | 0.79 | 3.9 | 0.08 | 0.91 |
| | Se P3 | 41 | 500 | 0.79 | 4.2 | 0.08 | 0.91 |
| | Se P4 | 45 | 500 | 0.69 | 3.8 | 0.15 | 0.84 |
| | Se P5 | 6 | 500 | 0.45 | 1.1 | 0.52 | 0.47 |
| Costa March 2016 | Se T1 | 26 | 500 | 0.58 | 2.7 | 0.24 | 0.75 |
| | Se T2 | 33 | 277 | 0.82 | 4.1 | 0.06 | 0.93 |
| | Se C1 | 27 | 504 | 0.54 | 2.5 | 0.27 | 0.72 |
| | Se C2 | 26 | 500 | 0.54 | 2.5 | 0.33 | 0.66 |
| | *Sargassum* T1 | 16 | 501 | 0.36 | 1.47 | 0.60 | 0.39 |
| | *Dyctiota* T2 | 14 | 502 | 0.15 | 0.60 | 0.85 | 0.14 |
| | *Jania* T2 | 10 | 500 | 0.26 | 0.88 | 0.71 | 0.28 |
| | *Padina* T2 | 31 | 500 | 0.76 | 3.78 | 0.10 | 0.89 |
| | *Sargassum* C1 | 3 | 501 | 0.28 | 0.44 | 0.84 | 0.15 |
| | *Dyctiota* C1 | 18 | 504 | 0.62 | 2.6 | 0.27 | 0.72 |
| | Alga 2 C2 | 17 | 500 | 0.47 | 1.92 | 0.47 | 0.52 |
| Control site March 2016 | Sed CS1 | 44 | 500 | 0.87 | 4.7 | 0.04 | 0.95 |
| | Sed CS2 | 53 | 501 | 0.77 | 4.4 | 0.08 | 0.91 |
| | Alga CS1 | 57 | 500 | 0.82 | 4.8 | 0.05 | 0.94 |
| | *Codium* CS1 | 20 | 504 | 0.33 | 1.4 | 0.62 | 0.37 |
| | *Sargassum* CS1 | 33 | 500 | 0.69 | 3.4 | 0.13 | 0.86 |
| | *Codium* CS2 | 23 | 519 | 0.24 | 1.1 | 0.71 | 0.28 |
| | *Gracilaria* CS2 | 19 | 502 | 0.27 | 1.1 | 0.70 | 0.29 |
| | *Sargassum* CS2 | 25 | 500 | 0.62 | 2.8 | 0.23 | 0.76 |

At the Coast zones (Tanques and Cuevas), S values of 26 and 33 taxa were recorded per sediment sample (N = 500). Computed diversity (H′) values ranged from 2.5 to 4.1, and relative low dominance values (λ = 0.06–0.27). Only station 2 (Tanques) yielded a low N = 277, albeit, with a H′ = 4.1. However, the epiphytic diatom assemblages had lower S values with 3 to 31 taxa per sample, than the sediments and macroalgae of the CS, where 19 to 57 diatom taxa were recorded (Table 5).

In particular, diversity values for the epiphytic diatoms on *Dyctiota* sp. and *Jania* sp. of station 2 in Tanques and on *Sargassum* sp. in Cuevas were very low, H′ = 0.60, 0.88 and 0.44, respectively, with corresponding high dominance values (λ= 0.85, 0.71 y 0.84). The diatom taxocoenosis on *Padina* sp. was exceptionally high in terms of species richness (S = 31 taxa) and diversity (H′ = 3.7). On *Sargassum* sp. only three taxa were observed, with *Cocconeis scutellum* reaching an abundance of 459 valves (N = 500). On *Jania* sp. at the Tanques zone 10 diatom taxa were recorded, with *G. aestuarii* reaching an abundance of 418 valve and *L. abbreviata* 62 valves. Other diatom taxa were rare (1–7 valves). On *Dyctiota* sp., *C. scutellum* reached 464 valves, very similar to *Sargassum* sp. (Table 5).

*4.5. Control Site Diatoms*

Species richness per sample at the CS ranged from 44 to 53 taxa, with corresponding high diversity values (H′ = 4.7 and H′ = 4.4.); in all substrates the N = 500 was reached (Table 5). On the inspected macroalgae 19 to 57 diatom taxa were recorded, corresponding to diversity values of H′ = 1.1 to 4.8. Macroalgae with low diversity of epiphytic diatoms

were *Codium* sp. from station 1 (H′ = 1.4), *Codium* sp. station 2 (H′ = 1.1), and *Gracilaria* sp. station 2 (H′ = 1.1), that corresponded with high dominance values λ = 0.62 and 0.71. These reflected the high abundance of *Licmophora flabellata* (N = 397, 439 and 420, respectively).

The values depicting the structure of benthic diatom assemblages from Port, Costa and CS (Table 5) showed no significant differences between them: Species richness (K-W = 3.89, d.f. = 2, p = 0.1425); H′ diversity (K-W = 3.5919, d.f. = 2, p = 0.166); Simpson's dominance (K-W = 4.2813, d.f. = 2, p = 0.1176), Equitability (K-W = 3.9064, d.f. = 2, p = 0.1418) or Simpson's diversity (K-W = 4.2819, d.f. = 2, p = 0.1176). Thus, the Ho cannot be rejected.

When using diatom species relative abundances, overall similarity between SR and CS reached 13.8%, which indicates that the epiphytic diatom association structure depends on quite distinct taxa at each locality. Although 115 taxa, roughly a fourth out of the 476 total taxa, were shared between them, and more than 50% of the SR species richness (215). The Bray–Curtis similarity dendrogram (Figure 4) shows a group formed by the SR Port diatom assemblages; a second group including diatom taxa from algae and sediment of Tanques and Cuevas from the SR coast and the CS, albeit at a 10% similarity value. However, at the 60% stablished similarity standard, no groups are recognized, except for small discrete pairings between epiphytic assemblages.

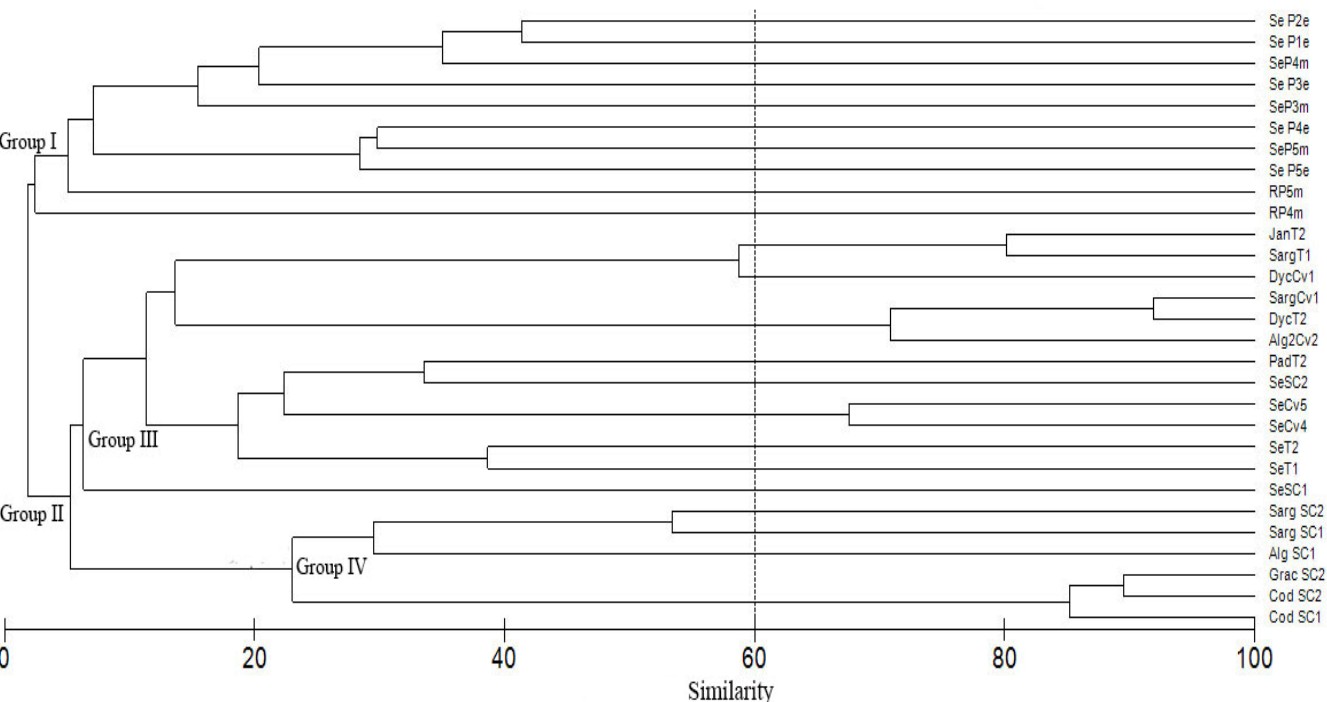

**Figure 4.** Bray-Curtis similarity between benthic diatom associations from SR and CS using relative abundances of the diatom taxa. P = Port, T = Tanques, Cv = Cuevas, CS = control site, R = rock, Se = sediment, Jan = *Jania*, Sarg = *Sargassum*, Dyc = *Dictyota*, Pad = *Padina*, Grac = *Gracilaria*, Cod = *Codium*.

As with relative abundances, presence-absence data for the benthic diatom taxa the computed values of similarity were very low, and the depicted groups are below the stablished similarity standard of 60% (Figure 3). This indicates that the species composition of the benthic diatom assemblages from Puerto, Costa and CS are different, but also between the substrates within the localities. This is highlighted in sampling point for sediments at the Port (SeP5e) where 6 taxa (*Amphora ocellata, Cocconeis disculoides, Fallacia litoricola, Nitzschia sigma, Staurophora salina, Thalassiosira* cf. *eccentrica*), which were very rare at the CS, Tanques, or Cuevas, except for *Staurophora salina* which was common in sediments at other stations.

A higher frequency of deformed valves was recorded from rocks and sediments of the SR coast. These belonged to *Achnanthes*, *Cocconeis*, *Diploneis*, *Navicula*, *Staurophora*, and *Thalassiosira*, and were more common for *Achnanthes* reaching from 1 to 7.6% (Table 6). In particular, *Achnanthes longipes* from a rock in station 5, out of an abundance of 115 valves, 23 were deformed (19.65%). In sediments from station 2, out of 56 specimens of *Achnanthes* sp., 8 deformed valves were observed (14.28%). Said deformities consist on the margin of the valve bending towards the axial region, deformed raphes, and less frequently, areolae alterations (Figure 5).

**Table 6.** Proportions of deformed valves of diatoms collected at Port and Costa (SR), and CS. DV = deformed valves, RA = relative abundances, M = May, J = January, Se = Sediment, R = rock, # = sampling station.

| SiteStation | Substrate | DV | RA | % |
|---|---|---|---|---|
| Port SR (polluted) | MSe3 | 2 | 169 | 1.1% |
| | Mse4 | 16 | 500 | 3.2% |
| | Mse5 | 6 | 500 | 1.2% |
| | MR5 | 38 | 500 | **7.6%** |
| | Jse1 | 4 | 500 | 0.8% |
| | Jse2 | 8 | 500 | 1.6% |
| | Jse3 | 5 | 500 | 1.0% |
| | Jse4 | 6 | 500 | 1.2% |
| | Jse5 | 1 | 500 | 0.2% |
| Costa SR (polluted) | *Sarg* T1 | 3 | 501 | 0.6 |
| | *Sed* Tan 1 | 2 | 500 | 0.4 |
| | *Sarg* C1 | 2 | 501 | 0.4 |
| | *Dyct* C1 | 2 | 504 | 0.4 |
| CS Non-polluted | Sed 1 | 1 | 500 | 0.2 |

At Costa (SR) only 9 deformed valves were observed out of 5289 from sediments and macroalgal substrates (0.17%). These belonged to *Licmophora gracilis* (1) and *Cocconeis scutellum* (1) on *Sargassum* sp. from station 1 in Tanques (Table 7). At station 2 in Cuevas two deformed valves of *Cocconeis scutellum* were collected from *Sargassum* sp., and on *Dyctiota* sp. *C. scutellum* (1) and *Gomphoseptatum aestuarii* (1) (Cuevas station 1), whilst in sediments of station 1, Tanques, deformed valves of *Navicula directa* (2) and *Navicula diversistriata* (1) occurred. From the CS, a single deformed valve of *Cocconeis* sp. was observed (Table 7; Figure 5).

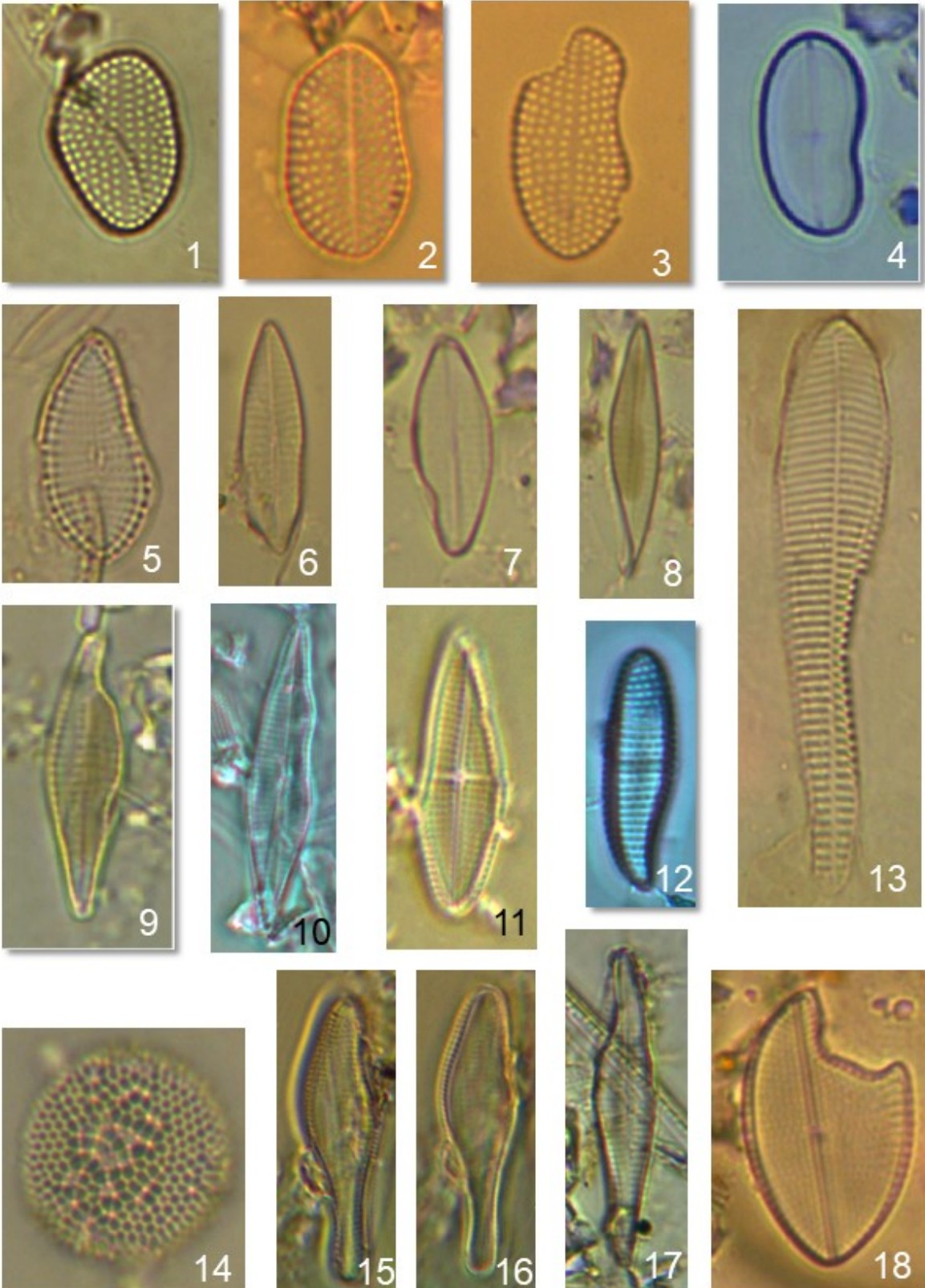

**Figure 5.** Deformed benthic diatom valves and areolae. (1)–(4) *Cocconeis* sp., (5) *Cocconeis* sp. (2), (6)–(8) *Navicula* sp., (9)–(12) *Achnanthes* sp. (13) *Licmophora gracilis* (Costa, SR), (14) *Thalassiosira* sp. (15)–(17) *Achnanthes* sp. (2), (18) *Cocconeis* sp. (sediment, CS). SR = Santa Rosalia, CS = control site.

**Table 7.** Deformed valves from the sampling stations at Port and Costa of Santa Rosalia, and Control Site (CS). M = may, J = January, Se = sediment, R = rock, T = tanques, C = cuevas.

| | Mse3 | Mse4 | Mse5 | MR4 | MR5 | Jse1 | Jse2 | Jse3 | Jse4 | Jse5 | Sarg T1 | Sarg C1 | Dyct C1 | Sed T 1 | Sed CS1 |
|---|---|---|---|---|---|---|---|---|---|---|---|---|---|---|---|
| *Achnanthes javanica* | | | | | 7 | | | | | | | | | | |
| *A. longipes* | | | | | 23 | | | | | | | | | | |
| *A. parvula* | | | | | 8 | | | | | | | | | | |
| *Achnanthes* sp. | 2 | 11 | | | | 2 | 8 | 4 | 1 | | | | | | |
| *A. yaquinensis* | | | | | | | | | 2 | | | | | | |
| *Cocconeis* sp. | | 1 | | | | | | | | | | | | | |
| *Diploneis litoralis* | | 1 | | | | 2 | | | | | | | | | |
| *Navicula subinflatoides* | | | 5 | | | | | | | | | | | | |
| *Staurophora salina* | | | 1 | | | | | 1 | | | | | | | |
| *Thalassiosira eccentrica* | | | | | | | | | 1 | | | | | | |
| *Shionodiscus oestrupii* | | 3 | | | | | | | 2 | 1 | | | | | |
| *Cocconeis scutellum* | | | | | | | | | | | 1 | 2 | 1 | | |
| *Gomphoseptatum aestuarii* | | | | | | | | | | | | | 1 | | |
| *Licmophora gracilis* | | | | | | | | | | | 1 | | | | |
| *Navicula directa* | | | | | | | | | | | | | | 2 | |
| *Navicula diversistriata* | | | | | | | | | | | | | | 1 | |
| *Coconeis* sp. | | | | | | | | | | | | | | | 1 |
| Total | 2 | 16 | 6 | 0 | 38 | 4 | 8 | 5 | 6 | 1 | 2 | 2 | 2 | 3 | 1 |

## 5. Discussion

### 5.1. Metals in Sediments

Both the port and coast of Santa Rosalia have been subject to impact by the mining activity of copper ever since 1868, with a brief closure in 1938 [50], and re-opening in 2014. During the first 70-year period mining wastes were poured into the adjacent sea, carried by water runoff, or by winds, with the consequent contamination of beaches and port. Geochemical studies have deemed this a highly contaminated area, yielding enrichment values for several metals, including Cu > Zn > Co > In > Cd > Mn > U [25–27,29,31] which remain high according to our current results. These indicate that the highest enriched element is copper, which is the main extracted resource in the area. Enriched concentrations for this metal surpass the standard value NEF of 50, reasserting the contaminated status of the area according to this criterion.

The above requires that the impact on the resident biota be assessed. For this we used as reference the standardized values for biota affectation at low range effect (LRE) and medium range effect (MRE) proposed by Buchman [51]. However, under laboratory conditions, concentrations at which affectation occurs are lower than the standards for various organisms, with a decrease in growth rate appearing at 26 mg kg$^{-1}$ Cu, 79 mg kg$^{-1}$ of Cd and 29 mg kg$^{-1}$ of Pb. Moreover, few related studies on marine benthic diatoms have recorded metal concentrations in polluted environments that surpass the upper earth cortex value (UCC), such as Cu, Pb, Zn and Cd, which are also higher than LRE values [7,22,52]. Values in SR are much higher, thus the response of benthic diatoms would seem obvious. However, in general, they are not.

### 5.2. Benthic Diatoms

Taxonomic composition of the benthic diatoms surveyed at the SR sampling stations, including both genera and species categories, coincide with those recorded in other polluted marine environments: *Nitzschia, Navicula, Cocconeis, Amphora,* and *Achnanthes* [7,8,20–24]. However, these are also the most widely distributed in most marine habitats. In particular, for both coasts of the Baja California Peninsula, most species reported belong to *Navicula, Nitzschia, Amphora, Mastogloia, Diploneis* and *Cocconeis* [53], that are the most common taxa in undisturbed ecosystems. Both the contaminated area (SR) and the non-polluted CS shared the same taxa: *Amphora* (28 taxa for SR and 43 for CS), *Nitzschia* (SR = 25, CS = 29), *Cocconeis* (SR = 19, CS = 31), *Diploneis* (SR = 14, C S = 28), *Navicula* (SR = 15, CS = 30), *Achnanthes* (SR = 11, CS 16), albeit, with noticeable difference in the number of species within each genus. This is reflected in the overall S for each locality, indicating that the contaminated environment (SR) harbors a much lesser species richness of benthic diatoms.

The observed differences in certain taxa from Puerto, Cuevas and the CS may be explained by the opportunistic growth of benthic diatoms due to particular site conditions and not to pollution. Nonetheless, given the opportunistic nature of benthic diatoms and their ubiquitous distribution taxonomic resolution in this type of research should be as precise as possible to allow comparison of different areas subject to any kind of contamination. In this case, however, the different composition of species observed between SR and the CS does seem to be caused by the contamination of metals in the former, which may also be precluding the growth of a diverse macroalgal substrate, consequently limiting the diversity of epiphytic diatoms, and explaining why the species richness at the control site (S = 397 taxa) nearly doubled that of SR (S = 217 taxa). Moreover, the 115 taxa shared by these localities are common all around the Mexican NW, although, earlier, three diatom taxa from SR were recorded for the first time in the region [34]. Meanwhile, 18 taxa identified at the CS may be deemed also as new records for the Baja California Peninsula. Said differences, however, are common and may be exemplified by the fact that similarity values between subsamples of the same sample commonly variate between 60 and 80% and denote the patchy distribution of diatom assemblages [3]. Notwithstanding, the taxonomy of the diatom assemblages from the control site adds relatively few taxa to the floristics

of the Baja California Peninsula coasts, while the number and composition of species are similar to those from recent studies in the area [45,54], while showing a typical structure. This is why species composition has to be coupled with numerical presence of said taxa to infer a given response to the contaminants in the sediments of SR. Thus, the observed departure of the polluted area of SR refers to the structure and species composition of benthic diatom assemblages in general [4].

Elsewhere, Potapova et al. [24] recorded *N. gregaria, Cyclotella atomus, C. marina* and *Nitzschia* sp. as the more common species in a zone contaminated by high concentrations of metals, whereasin SR, *Staurophora salina, A. ocellata, N. subinflatoides, Psammodyction panduriforme, A. javanica, Cocconeis scutellum* and *Gomphoseptatum aestuarii*, were the most abundant. Thus, biogeographical issues come into place that condition the species composition of benthic diatom assemblages that tolerate or resist this or other type of contamination.

According to the above, a numerical approach to the floristic use of organisms for assessing contamination was in order. So, considering that the number of species in a sample from marine habitats may range from 20 to 45 [4], several samples from SR meet this standard, while others such as sediment and rock from Port 3) do not (S = 6 and 8, respectively), where the highest concentrations for several metals in sediments were detected for (Cu = 7550, Zn = 4755, Ba > 5000, Sr = 7550, U = 110.5 mg kg$^{-1}$). It has to be underlined that also *Sargassum* sp. (C1, Cuevas) showed very low S, with three taxa.

Low S of benthic diatoms is observed under harsh conditions, such as those in hypersaline environments [18,55], although the natural patchy distribution of benthic diatoms may also be reflected in the number of taxa in a given sample. So, because contamination by metals deems an environment harsh, the high number of taxa in SR does not agree with the expected relation, except in two sampling points. Thus, the exceptional low abundances on the rock collected at sampling point 4, where only 14 valves from eight taxa were observed, but having high correspondent diversity values (H$'$ = 2.6) and dominance (λ = 0.13) should serve as a caveat when relating other structural attributes of the diatom assemblages (Table 5).

Moreover, diversity values in general at the Port stations (H$'$ = 2.4–4.3) are typical of benthic diatoms in undisturbed habitats. Except for station 5, where *N. subinflatoides* and *A. ocellata* were common, S and diversity were low (H$'$ = 1.9 y 1.1), and also in the macroalgal substrates, *Dyctiota, Jania* (Tanques 2) and *Sargassum* (Cuevas 1) showed very low values (H$'$= 0.60, 0.88 y 0.44, respectively), but with high values of dominance (λ = 0.60, 0.85, 0.84, and 1), thus favoring cuasi monospecific proliferations of *Cocconeis scutellum* and *Gomphonema aestuarii*, which are common epiphytic taxa on macroalgae. Earlier, ref. [43], had recorded S = 6 and H$'$ = 0.26 and a dominance of *G. aestuarii* in epiphytic diatom assemblages on *Ulva lactuca*, which was interpreted in terms of successional phases for an undisturbed habitat. These contrasting values may thus be depicting successional stages of benthic diatom assemblages which may decrease (under experimental conditions) from H$'$ = 4.33, down to 0.7, and later up to 3.5 [3]. The Kruskal–Wallis test shows that the values depicting the diatom assemblage structure for all sites (Port, Costa, CS) are not significantly different. Thus, those computed for the contaminated sites do not exhibit evidence of response to said metal pollution.

Likewise, the typical heterogeneity of benthic diatom distribution along various substrata was highlighted by the similarity analysis. Stations were grouped (although below the standard 60% line) by substrate, which is one of the factors that determine the patchy distribution of diatoms along environmental gradients, depicting different diatom associations in a same substratum and sampling site [3,43]; clouding the potential impact of a given contaminant agent. In this way, the observed benthic diatom associations on sediments, rocks, and macroalgae are not different from the typical ones that characterize those from undisturbed habitats [44]. Moreover, the following questions arise: What are the values of diversity in these stations during other seasons? And, to address successional variation, in shorter periods?

### 5.3. Element Contamination Comparison with Benthic Diatoms

For now, according to the above, the straightforward expectancy on the assemblage structure in the SR benthic diatom samples in response to the harsh conditions due to metal contamination is not met, and thus the Ho has to be accepted in part, albeit otherwise partially rejected on the basis of deformed valve frequency. Thus, a physiological response of benthic diatoms to contamination by metal should also be considered. Laboratory studies with marine diatoms have recorded their response in terms of mean lethal concentrations (EC50). In *N closterium* at Zn, Cd, Cu, Pb concentrations of (0.065 mg L$^{-1}$, 79 mg kg$^{-1}$, 26 mg kg$^{-1}$, 29 mg kg$^{-1}$) growth is inhibited (Stauber & Florence, 1990; Moreno-Garrido et al., 2003). Likewise, this effect has been observed in *Entomoneis* cf. *punctulata*, *Nitzschia* cf. *paleacea*, *Amphora coffeiformis*, *Odontella mobiliensis*, *Amphora hyalina*, *Thalassiosira nordenshkioelditii*, *Thalassiosira pseudonana*, and *Thalassiosira weissflogii*, with intracellular Cd accumulation [11–17]. Although the in vitro EC50 values are much lower than those measured in the Santa Rosalia polluted environment, there are also multiple factors interacting under in situ conditions which influence the potential effect of contaminating metals.

### 5.4. Deformed Frustules

A teratological response of diatom frustules as a physiological affectation has been deemed an indicator of metal contamination. In general, deformed valves are reported in very low frequencies. However, frequencies around 10% or higher are considered indicative of an impact of contamination by metals, while lower values may be normal [6,7,56], because, experimentally, a lineal correspondence of frustule deformities and an increase in concentration of Cd were not correlated below this percentage [57]. Teratological responses are frequent in nature and are attributable to certain levels of metal contamination which, on the basis of field observations, they proposed an arbitrary threshold of 7% of deformed valves of a given taxon, or 5% of the specimens in a sample [58]. Some studies have reported deformed valve frequencies of up to 15 a 23% for certain taxa, such as *Ulnaria ulna*, *Fragilaria* sp. and *Gomphonema parvulum*, with a high correlation between number of deformed valves and intracellular concentrations of metals [56,58].

In other marine studies dealing with contamination by metals no deformed frustules have been reported, except for Dickman (1998), who observed frequencies of 10% deformed valves of *F. capucina*, *A hauckiana*, *D. vulgare*, and *N. rhyncocephala* in stations with relatively high concentrations of Cu (400 mg kg$^{-1}$). Pb (130 mg kg$^{-1}$) and Zn (450 mg kg$^{-1}$) in sediment. Although most taxa were freshwater forms. In our study, a higher frequency of deformed valves was observed for *Achnanthes longipes* (14 y 19%) in the Port area (SR), with other taxa exhibiting lower frequencies in the Coast area (*L. gracilis*, *C. scutellum*, *G. aestuarii*, *N. directa*, *N. diversistriata*. However, in the CS, in all the abundant diatom samples but a single deformed valve was recorded. The data for SR may be considered a response to the high concentrations of several metals in the area, inasmuch 5% has been proposed as being caused by this type of contamination in rivers [56]. Thus, for the marine environments, the values corresponding to *A. longipes* may be considered a reliable reference to the response of benthic diatoms to metal contamination, branding this taxon as a tolerating species that lingers under presence of PTE.

## 6. Conclusions

The surveyed diatom taxocenosis at Santa Rosalia (Port) showed noticeable lower abundances and species richness, and a high frequency of deformed diatom valves, that strongly suggest a response to contamination by metals in spite of presenting a typical association structure for non-contaminated environments, which should be further analyzed in terms of its intrinsic characteristics and assessed as a reference for contamination. In the meantime, for the marine environments, the values corresponding to *A. longipes* may be considered a reliable reference to the response of benthic diatoms to metal contamination.

**Author Contributions:** The first author (Y.J.M.) conceived the original idea, participated in the sampling design, carried out the samplings and processed the samples, generated most of the data, and constructed the first *in extenso* report (PhD thesis document). The second author (D.A.S.-B.) theoretically contextualized the research, participated in the sampling design, diatom taxa identification, revised and interpreted the floristics and diversity values, and wrote the English version of the manuscript. The third author (A.J.M.-R.) was in charge of the metal concentration data generation, measurements and interpretation, and reviewing the manuscript. All authors have read and agreed to the published version of the manuscript.

**Funding:** This research received no external funding.

**Institutional Review Board Statement:** Not applicable.

**Informed Consent Statement:** Not applicable.

**Data Availability Statement:** Not applicable.

**Acknowledgments:** Metal concentrations measurements were validated at the National Research Council Canada Institute for National Measurement Standards. Alejandro Aldana and Alberto López Fuerte aided in field sampling. We acknowledge the critical comments by Víctor Cruz Escalona throughout the study. YJM received a doctorate scholarship from CONACyT. DASB and AJMR are COFAA and EDI fellows of the Instituto Politécnico Nacional. We thank also to Secretaría de Investigación y Posgrado, projects SIP-20171719, SIP20160972, SIP20170889, SIP20181382, SIP20195126, SIP20200709.

**Conflicts of Interest:** The authors declare no conflict of interest.

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
