# Peer review of "Response of Benthic Diatom Assemblages to Contamination by Metals in a Marine Environment"

_jmse, doi:10.3390/jmse9040443_

Round 1
Reviewer 1 Report
Please find my comments in the attached file.
Good luck with the revision!

Author Response
REPLY TO REVIEWER 1
The manuscript entitled „Responses of Benthic Diatom Assemblages to Contamination
by Metals in a Marine Environment“ is well written with clear outcomes. Still I would like to point out a few questions and suggestions and show some of tricky sections that I identified.
REVIEWER
The first issue is the choice of sampling sites, which was described in the manuscript and seems reasonable. I think the spatial variability of the diatom assemblages is very high, however there is only one control site (Santa Maria) which is used for comparison with the “polluted” sites. Do you think choosing more control sites would change the results? Why did you choose exactly this site and not multiple control sites along the coast for example?
AUTHOR’S REPLY
Although many other localities could be used adequately as control sites to be surveyed simultaneously, there are previous studies (surveys) that serve a reference as to the undisturbed structure shown by diatom taxocoenoses. We did such comparison. The use of several control sites is unpractical inasmuch it rquires hughe amount of work that may otherwise be directed toward a distiunct study. So the selection of our control site, as explained in the text, is based on the distance that ensured unpolluted conditions by mining activities as in Santa Rosalía.
REVIEWER
The second issue is the assumption that the metal pollution is the principal factor influencing the diatom assemblages. I would expect that especially in the port there will be many other pollutants affecting the local biota – such as hydrocarbons directly related to transportation, pollution from municipal waste water, or other types of pollution. Did you test some other environmental parameters apart metals?
AUTHORS REPLY
We did state a null hypothesis considering that diatom assemblages would not show said influence my metal pollution, in part because other factors, e.g. desiccation, salinity, waste water, and hydrocarbons as well may be also affecting them. However, the main factor are metal contaminants that have been studied and shown to be surpassing tolerance limits by living organisms. whilst, the other factors (except maybe hc) have identifiable effects we can filter in our analysis. This limitation is implied in the exploratory trait of our study.
REVIEWER
Do you think the location of sampling sites inside the port can affect the diversity? It seems the locality is somehow isolated from the open sea unlike the other sites including Tangue, Cuevas and control sites at Santa Maria. Can this feature restrict some diatom species to be found in the port? The sampling at each site was made in different phase of year (from May 2015 to March 2016). Is there some seasonal dynamics of diatom assemblages that may also explain the differences?
AUTHOR’S REPLY
As with the other questions, this is a legitimate one. But to be entirely comprehensive in a single study could compromise the feasibility of carrying it out. So this is a first approach that, besides providing a basis for further research questions, it permits the identifications of issues as these brought up by the reviewer, but that have to be undertaken separately.
REVIEWER
I have a few suggestions for the text itself:
I would appreciate some information on the influence of metals on diatoms already in the
Introduction (there is some information in the discussion section, but would be useful to have it in the introduction as well).
AUTHOR’S REPLY
Quite pertinent observation. Thus, the following paragraph was included: Besides the above, physiological responses of benthic diatoms to contamination by metal should are aslo an issue. In vitro studies using marine diatoms show their response to mean lethal concentrations (EC50). For example, the growth of N closterium at Zn, Cd, Cu, Pb concentrations of 0.065 mg L-1, 79 mg kg-1, 26 mg kg-1, 29 mg kg-1 is inhibited (Stauber & Florence, 1990; Moreno-Garrido et al., 2003). This effect has been observed also in Entomoneis cf. punctulata, Nitzschia cf. paleacea, Amphora coffeiformis, Odontella mobiliensis, Amphora hyalina, Thalassiosira nordenshkioelditii, Thalassiosira pseudonana, and Thalassiosira weissflogii, with intracellular Cd accumulation (Adams & Stauber, 2004; Franklin et al., 2001; Anantharj et al., 2011; Manimaran et al., 2012; French & Evans, 1988; Wang & Wang, 2011; Wang & Wang, 2009; Miao & Wang, 2006).
REVIEWER:
I would extend the conclusion section with some more generalized information. Is there some important outcome of the study that is applicable also to other sites contaminated by metals? For example some species that might be used as bioindicator of metal pollution or on contrary is there some species (or group of species) that are typically missing in contaminated environment?
AUTHOR’S REPLY
Also very pertinent. Thus we have included the following conclusion in this section and the abstract as well: In the mean time, for the marine environments, the values corresponding to A. longipes may be considered a reliable reference to the response of benthic diatoms to metal contamination.
REVIEWER
In general I am not completely convinced with the metals being the only factor influencing the diatom assemblages – it is probably an important factor, but others should be taken into account as well. I would expect at least to have the general overview of water properties at sampling sites (pH, temperature, oxygen) and basic chemical composition of the substrate for example.
AUTHOR’S REPLY
We agree, as has been mentioned above, but not as to the variables exposed by the reviewer; more important are dessiccation, salinity, fresh-water imput. This ecological issue is well documented and the responses of benthic diatoms to all are fairly known, but not the response to PTE which is what we are looking for, and that should deviate from the former.
REVIEWER
I think there might be a great potential of the study site (Santa Rosalia) to evaluate the impact of metal contamination when analysing the sediment cores. You stated that the mining activity occurred between 1868 and 1938, with consequent long closure and reopening in 2014. In case the metal concentration is affecting the diatom assemblages, this would be possible to find in the sediment record. You would have the “clean” environment before 1868, the polluted (1868-1938), recovery (1938-2014) and again polluted environment (recent). You would also obtain the precise metal concentrations in sediments of each period. This would avoid the problem with comparison of different localities with different other environmental factors influencing the resulting diatom assemblages.
AUTHOR’S REPLY
We could not agree more with the reviewer; if we later undertake this research question (the idea will be acknowleged to him). We would just have to hope that said cores are stable and show a good record of benthic diatoms.

Reviewer 2 Report
This is an article of sufficient quality to be published in JMSE.The objectives set, the methodology, the analysis of results and the conclusions are adequate.The influence of metal contamination on marine benthic diatoms is firmly established both by abundance and richness values, and by the different diatom species that define uncontaminated sampling areas as metal polluted ones. However, some aspects need to be corrected or improved.- Abstract: Do not use abbreviations that are explained later in the text (such as LRE or MRE). Write these texts without abbreviating.- Materials and methods- Results:- Table 3. Why is there no data for Al, Fe, Hg and Li in the control site? Have they not been detected or not measured?Review abbreviations: CS appears in the table and SC in the legend. BSC also appears in the legend but does not appear in the table.- Fig 2: Improve the legend where BCS is cited but in text it seems to be understood that the figure refers to the metals of Santa Rosalia (in general).- Fig. 3: The data refer only to control site. And in this case, there are values for Al, Fe, Hg and Li. Which is the reason? In any case, this figure is not necessary because in the text and in Table 3 there is enough information to conclude that the control is not contaminated by metals.- Table 4 is not considered in the discussion. See later.- Table 5. Three data are in bold letter. Why? It is not explained in the legend.- Table 6. The text does not reflect the data in the table. There is no data on the control (CS) in the table. For example, in the text it is said that “These reflected the high abundance of Licmophora flabellata (N = 397, 439 and 420, respectively) (Table 6)”, However, this species does not appear in the table.- Check numbering of tables from page 13. Table 5 must be table 7 and table 6 must be table 8. Also check numbering of figures (there is no fig. 5 but there is fig. 6)- Table 8. It is difficult to read. Improve formats.
- Discussion.
Lines 350, 351. It says: “few related studies on benthic diatoms have recorded metal concentrations in polluted environments”. But there are many studies of benthic diatoms exposed to high concentrations of metals in polluted rivers.
Lines 431-433. It says "the straightforward expectancy for low S and H ' values in the SR benthic diatom samples in response to the harsh conditions due to metal contamination is not met and thus the Ho has to be accepted”. The hypothesis cannot be accepted or rejected by applying only the criterion of the values of S or H '. There are other data in the study that allow rejecting Ho, such as the different species observed in each environment (table 4), or the groupings observed in the Bray-Curtis analyzes (Figs 4 and 5). S and H 'may be similar in different sampling areas but do not necessarily reflect the composition of diatom communities.In fact, the final conclusion of the study is controversial (lines 477-480) as it indicates that metal contamination affects the richness and abundance of diatoms. This last paragraph of this study should be revised. In general, the text should be carefully reviewed.- There are errors in the numbering of tables and figures.- There are errors in the abbreviations (SC or CS? Used in tables and figures.)- There are misspelled words: whreas (line 385), bellow (lines 214, 224), Mayo (line 327), etc.

Author Response
REVIEWER 2
However, some aspects need to be corrected or improved.
- Abstract: Do not use abbreviations that are explained later in the text (such as LRE or MRE).
Write these texts without abbreviating.
Agree, corrected in the MS
- Materials and methods
- Results:
- Table 3. Why is there no data for Al, Fe, Hg and Li in the control site? Have they not been detected or not measured?
These were added in the MS, it was an error updating the table.
Review abbreviations: CS appears in the table and SC in the legend.
Corrected in the MS
BSC also appears in the legend but does not appear in the table.
It was specified in the study area that BCS is Baja California Sur, the state to which Santa Rosalía belongs
- Fig 2: Improve the legend where BCS is cited but in text it seems to be understood that the figure refers to the metals of Santa Rosalia (in general).
Santa Rosalía belongs to BCS.
- Fig. 3: The data refer only to control site. And in this case, there are values for Al, Fe, Hg and Li. Which is the reason? In any case, this figure is not necessary because in the text and in Table 3 there is enough information to conclude that the control is not contaminated by metals.
We agree with the reviewer, hence we have removed figure 3 from the MS.
- Table 4 is not considered in the discussion. See later.
- Table 5. Three data are in bold letter. Why? It is not explained in the legend.
Were Corrected, no need for said highlighting
- Table 6. The text does not reflect the data in the table. There is no data on the control (CS) in the table. For example, in the text it is said that “These reflected the high abundance of Licmophora flabellata (N = 397, 439 and 420, respectively) (Table 6)”, However, this species does not appear in the table.
Licmophora flabellata is from the control site, and table 6 explains the Santa Rosalía species. It has already been updated in the text.
- Check numbering of tables from page 13. Table 5 must be table 7 and table 6 must be table 8.
Also check numbering of figures (there is no fig. 5 but there is fig. 6)
Yes, this has been corrected in the MS
- Table 8. It is difficult to read. Improve formats.
Modifications were done on Table 8 in compliance with the reviewer’s observation
REVIEWER
Lines 350, 351. It says: “few related studies on benthic diatoms have recorded metal
concentrations in polluted environments”. But there are many studies of benthic diatoms
exposed to high concentrations of metals in polluted rivers.
AUTHOR’S REPLY
We refer to studies in marine environments, thus we added the word "marine" in the text.
Lines 431-433. It says "the straightforward expectancy for low S and H ' values in the SR
benthic diatom samples in response to the harsh conditions due to metal contamination is not met and thus the Ho has to be accepted”. The hypothesis cannot be accepted or rejected by applying only the criterion of the values of S or H '. There are other data in the study that allow rejecting Ho, such as the different species observed in each environment (table 4), or the groupings observed in the Bray-Curtis analyzes (Figs 4 and 5). S and H 'may be similar in different sampling areas but do not necessarily reflect the composition of diatom communities. In fact, the final conclusion of the study is controversial (lines 477-480) as it indicates that metal contamination affects the richness and abundance of diatoms. This last paragraph of this study should be revised.
AUTHOR’S REPLY
To clear this, in the text we explain that we did not only resort to S y H´ but to the whole structure of the assemblages in order to reject the Ho. Plus, interpretation about Table 4 was added that explains that abundances in the different stations (port, Cuevas and CS) is caused by the opportunistic proliferation of diatoms.
In general, the text should be carefully reviewed.
- There are errors in the numbering of tables and figures.
These have been corrected
- There are errors in the abbreviations (SC or CS? Used in tables and figures.)
Also corrected in the MS
- There are misspelled words: whreas (line 385), bellow (lines 214, 224), Mayo (line 327), etc.
Also corrected

Reviewer 3 Report
The authors analyzed the composition of the diatom communities in areas affected by different degree of pollution by metals in order to determine their suitability as pollution indicators. Different attributes of the communities (abundance, taxonomical composition, diversity, percentage of deformed valves) growing on different substrata were determined. The authors concluded that abundance, biodiversity and deformed valves frequency differ in the polluted area compared to the clean area, indicating that these community attributes would be useful to assess the metal pollution in the marine environment. I think that the manuscript is of interest; the authors have done a valuable analysis work and the results are fairly useful; the analytical methods are sounds and some sections of the manuscript are well written. However, I also think that the manuscript maybe improved in several points; therefore, I find that it is not useful for publication in its present form but I encourage the authors to deal with the suggested changes. My main concerns are:
-The samples were collected at different dates and places. I guess that it implies that environmental variability plays a given role in explaining differences in the communities irrespectively of the pollution. Consequently, some data about the most relevant environmental variables that can affect the growth of diatoms occurring during the samplings should be included (e.g. temperature, salinity, nutrient concentrations).
-The work lacks of statistical analyses apart from the cluster analyses shown in Fig. 4 and 6 (which should be described in Material and Methods). I recommend the authors to use some ordination test (NMDS) to research and visualize the groupings of their samples according to the community structure. Additionally, the differences between places and substrata should be tested by using ANOVA or any other equivalent test (e.g. Kruskal-Wallis). Otherwise, the conclusions appear to be weak.
-The presentation of results should be improved. In particular, some of the Tables will be more easily visualized in graphs. Additionally, I found sentences confused that require to be rewritten (below is a non-exhaustive list). Therefore, a throughout revision of the text is necessary.
Specific comments:
-P1, lines 32-35. Please, to note that the species Latin name should be written in italic.
-P2. Line 73. The sentence containing 'on the basis of certainty that these represent real' is confused.
-P2. Line 85. The meaning of 'although having normal human settlements' is unclear.
-P3. Line 102. It is unclear what 'association structure' means in this context.
-P3. Lines 117-126. Please, to explain if the toxicity levels used by the mentioned authors in this paragraph are the same ones that are used in the present work (P5, line 177).
-P3. Line 133-135. See my previous comment about providing with some environmental data characterizing the sampling sites.
-P5 Line 177. The applicability of these toxicity values for the study area should be justified briefly.
-P6. Line 190. I am confused. Did the authors compared their results with the LRE and MRE values reported by Bouchman (2008)? Then, I wonder why other toxicity values are mentioned in P5, line 177 (Long et al. (1995).
-P6. Line 206. I guess that 'SC' in the legend of the Table should be 'CS'.
-P7,8. Figure 2 and 3. I think that these two Figures should have a similar format (i.e. the same variable on x- and y-axes). Additionally, in Figure 3 only selected metals in Fig2 should be shown. Please, to indicate in the legend what the different deviation measurements shown on the bars are.
-P8, lines 230-233. This sentence is confused.
-P9 Line 242. It is unclear what 'relative abundances' means. Please, to explain it clearly in Material and Methods. Additionally, I wonder if the differences in the diatom gathering procedures from different substrata (P4, lines 146-155) implies that the abundances obtained are not fully comparable.
-P9. Lines 250. Consider to use 'stations' instead of 'points'.
-P9. Lines250-252. This result illustrates that the diversity index has poor practical utility.
-P10. Table 5. I wonder why some values in H' column are highlighted in bold. Additionally, I think that the data shown in this Table might be shown in a Figure to do this part easier to be followed.
-P12. Figures 4 and 6 (note that Fig.5 is missed). I guess that these Figures are the result of a clustering analysis performed with the Bray-Curtis similarity matrix. The clustering technique used as well as the procedure for building the Bray-Curtis matrix should be detailed.
-P13. Table 6 might be presented as supplementary material.
-p16. Line 349. Please, mention which these organisms are.
-P16-17, lines 355-429. These are the reasons why a description of the environmental factors characterizing the sampled habitats is necessary to assess if taxonomic composition is a community attribute suitable for evaluating pollution.
Author Response
REVIEWER 3
The authors analyzed the composition of the diatom communities in areas affected by different degree of pollution by metals in order to determine their suitability as pollution indicators.
AUTHOR’S REPLY
That is not the aim of our study; it was clearly stated that we wanted to test the Ho that the benthic diatom assemblages would not show an effect due to the presence of PTE. This misunderstanding is not justified and may prejudice the methodology critic.
REVIEWER
Different attributes of the communities (abundance, taxonomical composition, diversity, percentage of deformed valves) growing on different substrata were determined. The authors concluded that abundance, biodiversity and deformed valves frequency differ in the polluted area compared to the clean area, indicating that these community attributes would be useful to assess the metal pollution in the marine environment.
AUTHOR’S REPLY
Not indicating but suggesting
REVIEWER
I think that the manuscript is of interest; the authors have done a valuable analysis work and the results are fairly useful; the analytical methods are sounds and some sections of the manuscript are well written. However, I also think that the manuscript maybe improved in several points; therefore, I find that it is not useful for publication in its present form but I encourage the authors to deal with the suggested changes. My main concerns are:
-The samples were collected at different dates and places. I guess that it implies that environmental variability plays a given role in explaining differences in the communities irrespectively of the
pollution. Consequently, some data about the most relevant environmental variables that can affect the growth of diatoms occurring during the samplings should be included (e.g. temperature, salinity, nutrient concentrations).
AUTHOR’S REPLY
That type of differences would not be of the same nature as those caused by pollution impact, and are well referentiated; variations due to the mentioned variables are well understood; we aimed to observe deviations from said differences.
-The work lacks of statistical analyses apart from the cluster analyses shown in Fig. 4 and 6 (which should be described in Material and Methods).
AUTHOR’S REPLY
We did not use statistical tests because there were no tests proposed at this exploratory stage, and because there is no need to statistically prove that metal concentrations values are much higher than the standard reference values. However, the basic statistical techniques required were used (definition of Statistics) and the (valid) inductive reasoning is there.
I recommend the authors to use some ordination test (NMDS) to research and visualize the groupings of their samples according to the community structure.
AUTHOR’S REPLY
We can not just insert such techniques; these should derive from a priori sampling design based on a precsie specific problem question. We are not here yet.
Additionally, the differences between places and substrata should be tested by using ANOVA or any other equivalent test (e.g. Kruskal-Wallis).
Not really, we are not making such comparisons for an statistical test to be required (still in inductive phase). Although We consider valid to use the suggested differences in order to design a statistical set of adequate tests for later comparisons (define objectives first).
Otherwise, the conclusions appear to be weak.
Not weak, they are hypothetical (post-facto) and are ready for the next step, which the reviewer is getting ahead of.
REVIEWER
-The presentation of results should be improved. In particular, some of the Tables will be more easily visualized in graphs.
AUTHOR’S REPLY
On this we disagree. Difficult to argument against an opinion based on individual perception
Additionally, I found sentences confused that require to berewritten (below is a non-exhaustive list). Therefore, a throughout revision of the text is necessary.
AUTHOR’S REPLY
We find these exagerated expressions uncalled for, inasmuch we and other reviewers do not see it as such. It is not ethical to try to magnify the importance of small errors, v.g. “not in italics” is a dactylographic error. Some of the following claims appear arrogant, i.e., if I do not understand it… it is not well written?
Specific comments:
-P1, lines 32-35. Please, to note that the species Latin name should be written in italic.
But...hey are in italics!
-P2. Line 73. The sentence containing 'on the basis of certainty that these represent real' is confused.
Because it is based on inspection of what actually occurs in nature, WHY CONFUSING?
-P2. Line 85. The meaning of 'although having normal human settlements' is unclear.
Not really, but maybe a: , and was missing. Corrected in the text
-P3. Line 102. It is unclear what 'association structure' means in this context.
We rephrased the sentence. However, the concept of structure was opportunely stated
-P3. Lines 117-126. Please, to explain if the toxicity levels used by the mentioned authors in this paragraph are the same ones that are used in the present work (P5, line 177).
We do not understand this questioning. Reference values are standrad international values
-P3. Line 133-135. See my previous comment about providing with some environmental data
characterizing the sampling sites.
Noted observation but unable to comply with. As we explained such variables have no bearing on the objective of our study.
-P5 Line 177. The applicability of these toxicity values for the study area should be justified briefly.
We disagree, inasmuch it has no bearing on the purpose of our reasearch, and other pertinent studies, some of which we cited have already done so.
-P6. Line 190. I am confused. Did the authors comparedtheir results with the LRE and MRE values reported by Bouchman (2008)? Then, I wonder why other toxicity values are mentioned in
P5, line 177 (Long et al. (1995).
Right, the second citation is the correct one. We changed it and eliminated the wrong reference
-P6. Line 206. I guess that 'SC' in the legend of the Table should be 'CS'.
Right, corrected
-P7,8. Figure 2 and 3. I think that these two Figures should have a similar format (i.e. the same variable on x- and y-axes).
Additionally, in Figure 3 only selected metals in Fig2 should be shown. Please, to indicate in the legend what the different deviation measurements shown on the bars are.
Also in agreement with R2, figure 3 was removed. And standard deviation values were incorporated.
-P8, lines 230-233. This sentence is confused.
Unable to comply
-P9 Line 242. It is unclear what 'relative abundances' means. Please, to explain it clearly in Material and Methods.
This part was corrected appropiately, indicating that relative abundance to N=500
Additionally, I wonder if the differences in the diatom gathering procedures from
different substrata (P4, lines 146-155) implies that the abundances obtained are not fully comparable.
This is a proven standard method for comparing benthic diatom assembalges within and among substrates
-P9. Lines 250. Consider to use 'stations' instead of 'points'.
Agree
-P9. Lines250-252. This result illustrates that the diversity index has poor practical utility.
Not the objective of this study to state this, and without sufficient basis.
-P10. Table 5. I wonder why some values in H' column are highlighted in bold. Additionally, I think that the data shown in this Table might be shown in a Figure to do this part easier to be followed.
Highligh removed. We do not agree on using a figure
-P12. Figures 4 and 6 (note that Fig.5 is missed). I guess that these Figures are the result of a clustering analysis performed with the Bray-Curtis similarity matrix. The clustering technique used as
well as the procedure for building the Bray-Curtis matrix should be detailed.
The required explanation was included in Material and Methods
-P13. Table 6 might be presented as supplementary material.
We choose to keep it as it is
-p16. Line 349. Please, mention which these organisms are.
All relevant taxa are mentioned
-P16-17, lines 355-429. These are the reasons why a description of the environmental factors characterizing the sampled habitats is necessary to assess if taxonomic composition is a community
attribute suitable for evaluating pollution.
We have covered this. The ecology and distribution of benthic diatom taxa from the region serve as reference: structure of the taxocoenoses described in previous studies that present a structural pattern and roughly related to the traditional measurements of those environmental variables. This is related to the opportunistic nature of diatoms that, notwithstanding, show structures that do not vary, changing mainly in species composition according to the type of habitat, but that is distinguishable from an impacted one by pollution.

Round 2
Reviewer 1 Report
Review of the manuscript “RESPONSES OF BENTHIC DIATOM ASSEMBLAGES TO CONTAMINATION BY METALS IN A MARINE ENVIRONMENT“
Manuscript ID: jmse-1160829
Dear authors,
thanks for considering my suggestions and implementing the changes in the manuscript. I still have few minor remarks. I numbered them according to your cover letter/replies…
1/ I would add some explanation of why Santa Maria is the control site – I understand the location of the site in respect with the ocean currents and thus not being influenced by metal contamination, but still some information on why you consider the locality as representative of undisturbed environmental conditions of the region would be good. Are there some published papers from the region to help illustrate the representativeness of the locality? If I understand well, you did some more sampling in the region but the data are not yet published – you mentioned several times Martinez et al (unpublished) – I am not completely sure about the journal policy of referencing unpublished data, perhaps you can instead add some relevant data…
In the discussion (line 484) you basically reported, that the diatom assemblage at the control site is somehow different from what is usually found in the region of Baja California. “In contrast, 18 taxa identified at the CS may be deemed also as new records for the Baja California Peninsula.“ Then, the locality is quite disputable to be considered as a control site…
Please, try to explain this issue in the study site section or Methods…
2 (how important factor is the metal pollution in comparison with other environmental factors) and 3 (location of samples inside the port) - I didn’t mean to remake the analysis or paper itself, but it would be fair to mention these issues in the discussion section!
The other issues were resolved reasonably, thanks for it!
Good luck with that, regards,
Jan

Author Response
REVIEWER 1
1/ I would add some explanation of why Santa Maria is the control site – I understand the location of the site in respect with the ocean currents and thus not being influenced by metal contamination, but still some information on why you consider the locality as representative of undisturbed environmental conditions of the region would be good. Are there some published papers from the region to help illustrate the representativeness of the locality? If I understand well, you did some more sampling in the region but the data are not yet published – you mentioned several times Martinez et al (unpublished) – I am not completely sure about the journal policy of referencing unpublished data, perhaps you can instead add some relevant data...
In the discussion (line 484) you basically reported, that the diatom assemblage at the control site is somehow different from what is usually found in the region of Baja California. “In contrast, 18 taxa identified at the CS may be deemed also as new records for the Baja California Peninsula.“ Then, the locality is quite disputable to be considered as a control site...
Please, try to explain this issue in the study site section or Methods...
AUTHOR’S REPLY
IN AGREEMENT WITH THE REVIEWER’S OBSERVATION WE MADE THE FOLLOWING MODIFICATIONS IN:
METHODS
Also, sediments and macroalgae thalli were collected in two points at the control site. The control site (CS) was located 8 km northward of SR in the locality known as Santa Maria in order to avoid any metal pollution from the SR mining activities. There, pristine conditions ensure that the benthic diatom assemblages used as reference would warrant high species richness and a species composition related to previous surveys in the gulf region (Moreno et al., 1996; Siqueiros Beltrones, 2002) and those underway at the time (Martínez & Siqueiros-Beltrones, 2018; López Fuerte et al., 2019).
AND THE FOLLOWING TEXT WAS ADDED IN DISCUSSION
While, 18 taxa identified at the CS may be deemed also as new records for the Baja California Peninsula. Said differences, however, are common and may be examplified by the fact that similarity values between subsamples of the same sample commonly variate between 60 and 80% and denote the patchy distribution of diatom assembalges (Siqueiros Beltrones, 2002). Notwithstanding, the taxonomy of the diatom assemblages from the control site adds relatively few taxa to the floristics of the Baja California Peninsula coasts, while the number and composition of species are similar to those from recent studies in the area (López_Fuerte et al., 2010; López-Fuerte et al., 2020), while showing a typical structure. This is why species composition has to be coupled with numerical presence of said taxa to infer a given response to the contaminants in the sediments of SR.Thus, the observed departure of the polluted area of SR refers to the structure and species composition of benthic diatom assemblages in general (Siqueiros Beltrones, 2005).
REVIEWER
2 (how important factor is the metal pollution in comparison with other environmental factors) and 3 (location of samples inside the port) - I didn’t mean to remake the analysis or paper itself, but it would be fair to mention these issues in the discussion section!
AUTHOR’S REPLY
WE SYMPATHIZE WITH THE REVIEWER’S CONCERNS, BUT WE ARE CONFIDENT THAT THE FOLLOWING EXCERPTS FROM OUR TEXT COMPLY WITH THEM
(LINES <500)
Low S of benthic diatoms is observed under harsh conditions, such as those in hypersaline environments (Siqueiros-Beltrones, 1988; Siqueiros-Beltrones et al., 2015), although the natural patchy distribution of benthic diatoms (McIntire & Overton, 1971) may also be reflected in the number of taxa in a given sample. So, because contamination by metals deems an environment harsh, the high number of taxa in SR does not agree with the expected relation, except in two sampling points. Thus, the exceptional low abundances on the rock collected at sampling point 4, where only 14 valves from 8 taxa were observed, but having high correspondent diversity values (H´= 2.6) and dominance (λ= 0.13) should serve as a caveat when relating other structural attributes of the diatom assemblages (Table 5).
AND (LINES >500)
Likewise, the typical heterogeneity of benthic diatom distribution along various substrata was highlighted by the similarity analysis. Stations were grouped (although below the standard 60% line) by substrate, which is one of the factors that determine the patchy distribution of diatoms along environmental gradients (McIntire & Overton, 1971), depicting different diatom associations in a same substratum and sampling site (Siqueiros-Beltrones, 2002; Hernández-Almeida, 2009), clouding the potential impact of a given contaminant agent.
ON THE ISSUE ABOUT THE ASSEMBLAGES INSIDE THE PORT, WE FEEL THAT IT IS A LEGITIMATE QUESTION, BUT SHOULD BE ADDRESSED ALSO AS A NEW STUDY PROPOSAL ALONG WITH A PREVIOUS SUGGESTION BY THE REVIEWER.

Reviewer 3 Report
I am rather surprised by the reaction of the authors, whose tone I feel that is inappropriate and unfair, in particular that allusion to my personal ethics. As I mentioned in my previous report: “I think that the manuscript is of interest; the authors have done a valuable analysis work and the results are fairly useful; the analytical methods are sounds and some sections of the manuscript are well written. However, I also think that the manuscript maybe improved in several points”. Obviously, the objective of my comments (of course, debatable and affordable or not by the authors according their own criteria) was just to aid in improving the manuscript. Honestly, I have the feeling of having wasted time in some way.
Regardless of such an outburst, I am striving to be a positive and think that the manuscript has improved with the changes done by the authors. At any rate, I still recommend that the authors applying some statistical tests to support their “suggestions”. I cannot understand why a study in exploratory stage should not shrink back from using statistical test (so I know, it is a common practice in environmental sciences where variability of the data is relevant). In addition, I think (of course, personal opinion) that the interest of the manuscript will increase if some data of relevant environmental variables are added (even although they have no bearing on the objective of the study).
Author Response
REVIEWER 3
I am rather surprised by the reaction of the authors, whose tone I feel that is inappropriate and unfair, in particular that allusion to my personal ethics. As I mentioned in my previous report: “I think that
the manuscript is of interest; the authors have done a valuable analysis work and the results are fairly useful; the analytical methods are sounds and some sections of the manuscript are well written. However, I also think that the manuscript maybe improved in several points”. Obviously, the objective of my comments (of course, debatable and affordable or not by the authors according their own criteria) was just to aid in improving the manuscript. Honestly, I have the feeling of having wasted time in some way. Regardless of such an outburst, I am striving to be a positive and think that the manuscript has improved with the changes done by the authors.
AUTHOR’S REPLY
First of all, and to avoid unnecessary discussion to clear the above, We simply apologize and express our gratitude again for the effort in reviewing the MS.
REVIEWER 3
At any rate, I still recommend that the authors applying some statistical tests to support their “suggestions”. I cannot understand why a study in exploratory stage should not shrink back from using statistical test (so I know, it is a common practice in environmental sciences where variability of the data is relevant). In addition, I think (of course, personal opinion) that the interest of the manuscript will increase if some data of relevant environmental variables are added (even although they have no bearing on the...
AUTHOR’S REPLY
After given more thought to the reviewer observations, We admit that there is a reasonable doubt on the need to include a statistical test for the values depicting the structure of benthic diatom assemblages (good call).
However, we disagree on the the rest as we have argumented in the previuos cover-letter
Thus, we made the following additions to the manjuscript
METHODS
The diversity values from the derived matrix were compared for significance differences among the Puerto, and Costa (Tanques and Cuevas, SR), and CS stations, using a Kruskal-Wallis (one
way) test for the Ho=no signicant differences would be detected.
RESULTS
The values depicting the structure of benthic diatom assemblages from Port, Costa and CS (Table 5) showed no significant differences between them: Species richness (K-W=3.89, d.f.=2, p=0.1425); H´ diversity (K-W=3.5919, d.f.= 2, p=0.166); Simpson’s dominance (K-W=4.2813, d.f.=2, p=0.1176), Equitability (K-W=3.9064, d.f.=2, p=0.1418) or Simpson’s diversity (K-W=4.2819, d.f.= 2, p=0.1176). Thus, the Ho can not be rejected.
DISCUSSION
The Kruskal-Wallis test shows that the values depicting the diatom assemblage structure for all sites (Port, Costa, CS) are not significantly different. Thus, those computed for the contaminated sites do not exhibit evidences of response to said metal pollution.
SINCERELY
THE AUHTORS
